Review Article

# Modular architecture of K$^+$ channels: the functional plasticity of the pore module

Oliver Rauh [ID] [1,2], Tobias Schulze [ID] [1], James L Van Etten [3], Gerhard Thiel [ID] [1,4] & Anna Moroni [ID] [4✉]

## Abstract

Miniature K$^+$ channel proteins from viruses (Kcv) are structurally and functionally equivalent to the pore module of all K$^+$ channels. Here, we summarize data in support of the hypothesis that pores of primitive K$^+$ channels served as building blocks for evolving the modern complex mammalian ion channels. Experimental data show that mutations in Kcv channels can generate gating phenomena like slow-activating inward or outward rectification, which are typical of complex mammalian channels. Hence, the basic mechanism for rectification is an inherent property of the pore module, which was further tuned and/or amplified during evolution by the addition of sensory protein domains. This evolutionary trend can be experimentally mimicked by coupling small pore units with a voltage-sensing domain or a glutamate-binding domain to acquire voltage and ligand-sensitive gating. The same modularity principle can be exploited in the design of synthetic channels in which the Kcv pore is coupled to orthogonal sensor domains. These synthetic channels exhibit new gating properties like a sensitivity to light or Ca$^{2+}$, which originate from their attached sensor domains.

**Keywords** Modular Evolution; Synthetic Potassium Channels; K$^+$ Channel Pore Module; Kcv; Chlorella Viruses
**Subject Categories** Evolution & Ecology; Membranes & Trafficking

## Introduction

Sequence homologies as well as conserved structural topologies in K$^+$, Na$^+$ and Ca$^{2+}$ channels suggest several layers of modularity in their architecture. On a macroscopic scale the six transmembrane domain (6TMD) motif of the monomers of canonical voltage-gated K$^+$ channels (K$_V$ channels) are repeated four times in voltage-gated Na$^+$ and Ca$^{2+}$ channels (Fig. 1) (Nelson et al, 1999; Anderson and Greenberg, 2001). This structural kinship has led to the suggestion that the latter two channels have presumably evolved stepwise in the prokaryotic world via gene duplication and gene fusion of primordial K$_V$-type channels (Nelson et al, 1999; Anderson and Greenberg, 2001; Zakon, 2012).

Structural and functional scrutiny reveals a further level of modularity in the highly conserved 6TMD topology of K$_V$ channels (Choe, 2002). Each monomer is, according to structural and functional aspects, composed of three building blocks: the pore module, the voltage-sensing domain, and the cytosolic domains (Fig. 1). The central pore module of a K$_V$ channel is made from two C-terminal transmembrane helices called S5 and S6. The latter are connected via a loop, the P-loop, containing the typical K$^+$ channel selectivity filter sequence. The S5/S6 helices plus P-loop of the monomers are grouped around a water-filled pore, which serves as the ion-conducting pathway in the final tetrameric arrangement of a channel. This so-called pore module, which is, with minor deviations, shared by all K$^+$ channels (Fig. 1), determines their main functional features, namely K$^+$ selectivity and gating. The conserved structural elements in this pore-forming unit are illustrated in the inset of Fig. 1 as a cartoon representation of a typical K$^+$ channel pore. The figure depicts in a side-view orientation two of the four monomers, which form a functional channel pore around the central ion-conducting pathway. The main structural elements of these pore-forming monomers are highlighted. Note that the domains S5 and S6 from the 6TMD channels (Fig. 1) are equivalent to the transmembrane domains TMD1 and TMD2 in the stand-alone pore module (Fig. 1, inset).

The current view of K$^+$ channel function is that the pore module contains two main gates, one at the entrance between the cytosol and cavity (inner gate) and a second in the selectivity filter (filter gate) (Kopec et al, 2019). These gates are responsible for stochastic fluctuations between open and closed states, which are already present in simple pore-only channels built of not more than the pore module (Rauh et al, 2017, 2022; Santos et al, 2008).

The second building element, the voltage-sensing domain (VSD) of K$_V$ channels, is formed by the four N-terminal transmembrane helices (S1–S4) (Fig. 1). Functional and structural data support a model in which the voltage sensitivity of these channels can be attributed mostly to the S4 transmembrane domain (Loots and Isacoff, 2000; Bassetto et al, 2023). Multiple cationic amino acids in the S4 helix interact with anionic charges in the densely packed transmembrane segments S2 and S3. The remaining net positive charges in S4 sense the electrical field, and their movement within the electrical field is transmitted to the gates in the pore module. The required mechanical coupling between both modules occurs in channels with a so-called "swapped" architecture via a linker between S4 and S5 (Barros et al, 2019). The S4–S5 linker in these

[1]Department of Biology, TU-Darmstadt, Darmstadt, Germany. [2]Institute for Functional Gene Analytics, Department of Natural Sciences, Bonn-Rhein-Sieg University of Applied Sciences, Rheinbach, Germany. [3]Department of Plant Pathology & Nebraska Center for Virology, University of Nebraska-Lincoln, Lincoln, NE, USA. [4]Department of Biosciences, University of Milan, Milan, Italy. ✉E-mail: anna.moroni@unimi.it

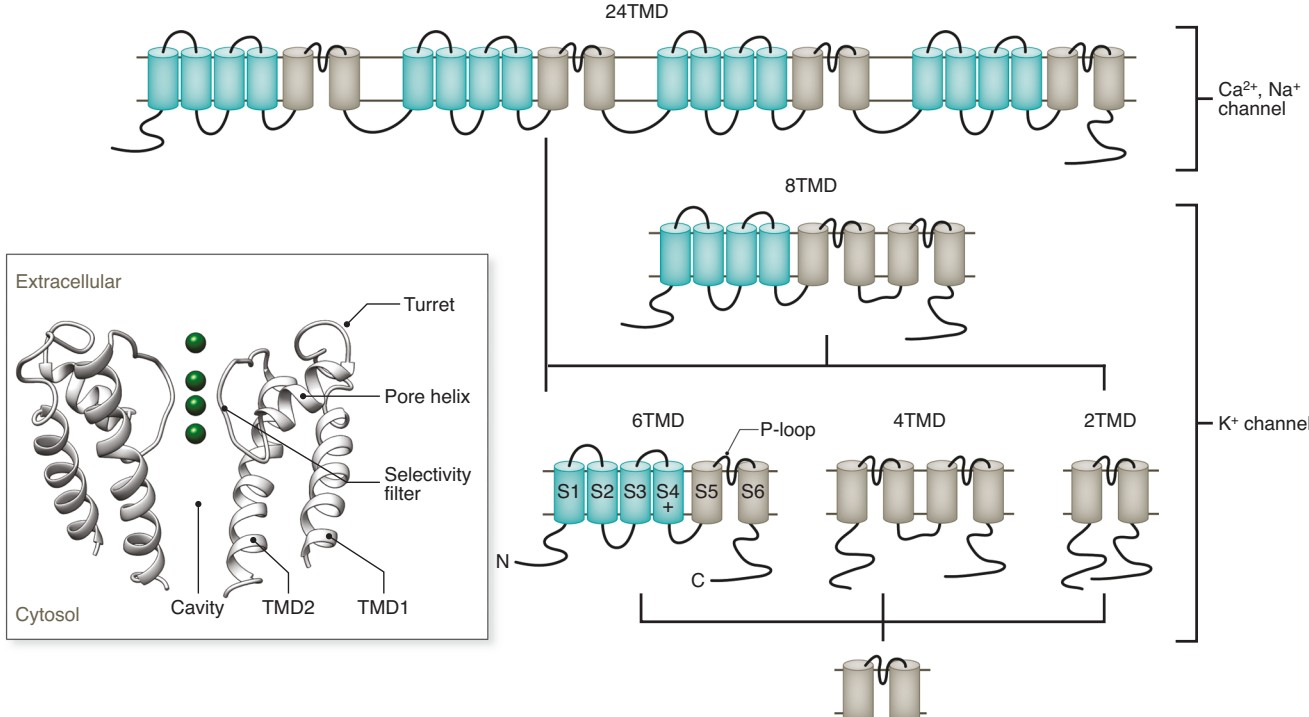

**Figure 1.    Modular architecture and hypothetical phylogenetic relationship between K$^+$, Ca$^{2+}$ and Na$^+$ channels.**

Schematic view on evolution of cation channels (from bottom to top). The presumed common ancestor of all K$^+$ channels and voltage-gated Ca$^{2+}$ and Na$^+$ channels is a prokaryotic or viral protein with two transmembrane domains (2TMD, in gray) with the structural hallmarks of the pore module of all K$^+$ channels. This 2TMD architecture is still maintained in modern Kir channels. Gene duplication of the 2TMD motif resulted in the 4TMD architecture represented by modern K2P channels. Fusion of a 2TMD ancestor channel with a membrane protein comprising 4TMDs (green) resulted in the conserved architecture of K$_V$ channels with a 6TMD build. A hybrid of the 6TMD and 2TMD architecture is found as 8TMD unit in yeast. Subsequent events of gene duplication and fusion of 6TMD units plus alterations in the cation selectivity presumably resulted in the formation of the first Ca$^{2+}$ and then Na$^+$ channels in which the 6TMD motif is repeated four times (24TMD). The nomenclature of transmembrane segments S1-S6, the presence of a cationic AA in S4 and the position of the pore loop are indicated in the 6TMD structure. Model based on Anderson and Greenberg, 2001. Inset: Structural elements of the pore-forming unit common to all K$^+$ channels are illustrated by a cartoon representation of a K$^+$ channel pore in side-view orientation with two of the four monomers. A pore module monomer is composed of an outer (TMD1) and an inner (TMD2) transmembrane domain. The two TMDs are connected via an unstructured extracellular turret domain, the α-helical pore domain as well as the selectivity filter. Four monomers are grouped around a central ion-conducting pathway, which is formed in series by the selectivity filter and the cavity; K$^+$ ions (green spheres) are depicted in the four filter binding sites.

channels is a rigid alpha helix that positions the S4 helix of one subunit near the S5 helix of the neighboring subunit. In the tetrameric channel, the S4–S5 linker helices are in this way positioned parallel to the membrane plane, wrapping the pore domain like a "cuff" (Long et al, 2007). In this way, a movement of the S4 helix in one monomer can be mechanically coupled via the adjacent monomer to the pore.

In contrast, in channels with a "non-swapped" arrangement, direct molecular contacts between the transmembrane helices S4 and S5 are more important for transmitting the position of the VSD to the pore (Barros et al, 2019). The S4–S5 linkers are in these channels only short loops instead of rigid helices. The S4 helix is in this way positioned close to the S5 of the same subunit. It has been shown that in this case, the linker is not required; channels maintained their voltage dependency even after truncating the covalent bond in the S4–S5 linker (Barros et al, 2019).

The third functional unit of K$_V$ channels is provided by the cytosolic domains (Figs. 1 and 2). These long stretches of soluble C- and N-terminal amino acids contain a variety of well-known binding sites (Barros et al, 2012). Their interactions with ligands serve as an additional layer of channel regulation in that they can

determine in a singular manner channel opening or closing. The cyclic nucleotide-gated ion channels (CNG channels) are an example for channels, which are directly activated by ligand binding to a cytosolic domain (Matulef and Zagotta, 2003). In other channels, ligand binding to cytosolic domains modulates channel activity in an allosteric manner. In the case of HCN channels, for example, cyclic nucleotides regulate allosterically the impact of voltage as a dominating gating factor (Wahl-Schott and Biel, 2009).

This apparent structural and functional modularity of K$_V$ channels has led to the hypothesis that they may have evolved by fusion of independent functional units to the common pore module (Patten et al, 1999; Nelson et al, 1999; Anderson and Greenberg, 2001). This idea is indirectly supported by a bulk of structural and functional evidence. One important piece in this puzzle was the finding of small pore-only K$^+$ channels in viruses, which are built of not more than the structural elements shown in the inset of Fig. 1 (Plugge et al, 2000; Gazzarrini et al, 2009). These channels are the main topic of this review and will be described below in more detail.

More support for the concept of modular evolution comes from a close look at the VSD as well as at cytosolic domains in K$_V$

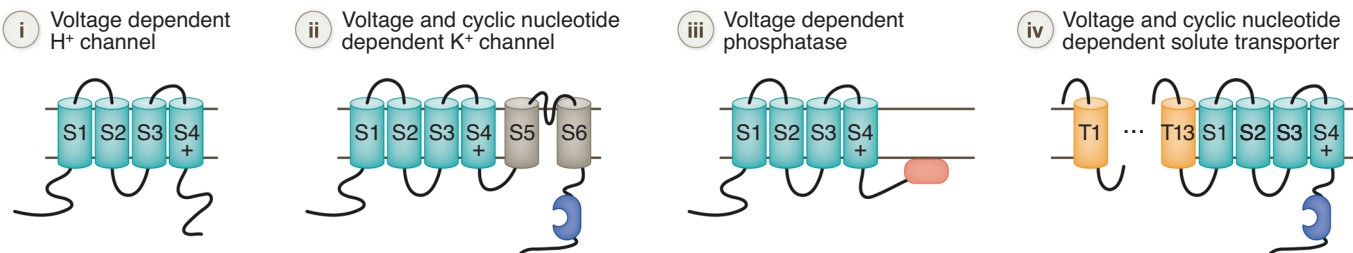

**Figure 2. Contextual occurrence of the voltage-sensing domain (VSD) in different proteins.**

The VSD is found as a stand-alone pore in a voltage-gated H⁺ channel H_V (i) and as a sensor domain in K_V channels (ii), a phosphatase (iii), or a solute transporter (iv). The sketch illustrates the general structure of the VSDs (green) with four transmembrane domains (S1–S4) and cation-rich S4 domain (+). The topology of the VSD as a modular element of complex proteins is shown with the pore domain (gray) of a K⁺ channel (ii, see Fig. 1), with a phosphatase (iii, red), which dephosphorylates phospholipids in the inner membrane leaflet and a solute transporter composed of 13 transmembrane domains (T1–T13, orange). The cytosolic N-terminus can in some proteins contain a CNBD (blue), which is linked directly to the VSD (iv) or alternatively to the pore domain (ii) as in the case of HCN channels.

channels. If complex K⁺ channels evolved from the fusion of individual building modules, one would expect to also find these building elements as stand-alone components or as functional modules in combination with other protein partners. In support of this prediction, protein domains with the same architecture of the VSD in K_V channels (Fig. 2ii) can be found in various organisms as stand-alone proteins (Fig. 2i) where they function in the form of dimeric proteins as H⁺ conducting channels (Ramsey et al, 2006; Sasaki et al, 2006). Also, in agreement with this view of a modular evolution is the finding that VSD-like domains serve as the membrane anchoring part of a cytosolic enzyme (Fig. 2iii). In the sea squirt *Ciona intestinalis* a VSD-like domain, named Ci-VSD, is part of a phosphatase (Murata et al, 2005). It is well established that changes in the electrical field elicit in K_V channels a minute movement of the charge-bearing VSDs. This small charge displacement perpendicular to the membrane plane can be measured as so-called "gating currents". It is most remarkable that the same kind of gating currents are also observed with the Ci-VSP (Okamura et al, 2009). It has been experimentally confirmed that the latter is a property of the voltage-sensing domain (Ci-VSD) and occurs also when separated from the phosphatase domain (Okamura et al, 2009). In the full-length protein, this displacement of the Ci-VSD in the electrical field imposes a voltage control over the activity of the phosphatase on the cytosolic side of the membrane.

In the context of modular evolution, it is worth mentioning that the voltage-sensitive phosphatase functions as a monomer (Kohout et al, 2008) meaning that the movement of a single Ci-VSD unit is sufficient for controlling the activity of the enzyme. K_V channels on the other hand have four VSDs and several of them are only opening after displacement of all four voltage sensors (Fedida and Hesketh, 2001). This suggests a common origin of voltage sensing in both types of proteins with a further evolutionary refinement in K_V channels.

Like the VSD, the cytosolic domains of ion channels contain functional modules, which are not unique to this type of protein. Regulatory domains such as the cyclic nucleotide-binding domain (CNBD) in HCN channels (Fig. 2), the **P**er-**A**rnt-**S**im (PAS) domain in HERG channels (Barros et al, 2012), or the KTN (**K**⁺ **t**ransport, **n**ucleotide binding) binding domain in various other K⁺ channels (Roosild et al, 2009) are also known as ligand binding domains for other enzymes, including kinases, phosphoesterases,

and transporters (Choe, 2002; Weber et al, 1987; Windler et al, 2018). As in the case of the VSDs, this suggests an evolutionary "repurposing" of effective binding motives for ligand-mediated regulation of different target proteins, including ion channels.

A very interesting example, which nicely illustrates the general principle and plasticity of a modular evolution is found in a voltage and cAMP-regulated solute transporter, called SLC9C1 (Windler et al, 2018). It contains, like some K_V channels (Fig. 2ii), a VSD- and a CNBD-like domain, which lend a voltage and cAMP-dependent regulation of this transporter protein. One topological peculiarity is that the CNBD is C-terminally attached to the VSD and not, like in HCN channels, to the channel pore domain. Even more interesting, however, is the modular topology of the VSD and the transporter domain: The VSD is in the solute transporter linked to the C-terminal end of the 13TMD protein and not (Fig. 2iv), like in K_V channels, to the N-terminus (Fig. 2ii). Recent cryoEM structures of the sea urchin SLC9C1 protein provide information on how the S4 domain in this peculiarly oriented VSD is in direct contact with the cytosolic CNBD and how a combination of cAMP binding and negative voltage releases an autoinhibitory mechanism in this transporter (Yeo et al, 2023).

The modular architecture of K_V channels, which is apparently composed of several distinct and independent modules, suggests a mosaic type of evolution. Complex modern K⁺ channels may have in this way enriched their functional diversity by the acquisition of sensory domains to the pore. This attractive hypothesis has several implications, which can be experimentally tested. The proposed modular evolution of K⁺ channels predicts:

(i)   An isolated pore module of K⁺ channels with intrinsic gates should be functional and should have existed in nature.
(ii)  It should be possible to reconstruct gating phenotypes of complex K⁺ channels in the isolated pore.
(iii) It should be possible to rebuild complex K⁺ channels with their respective functional properties by combining the pore module with peripheral sensory domains from evolutionary unrelated sources.
(iv)  Finally, it should be possible to engineer channels with new functional properties, which are not present in nature, by combining a channel pore module with orthogonal sensing domains.

Here we summarize recent data which experimentally confirm the above predictions for a modular evolution of complex K[+] channels. In the center of the review are miniature-sized K[+] channel proteins from viruses. They resemble from a structural and functional perspective, the conserved pore module of complex K[+] channels. The structural and functional diversity of these small channel proteins mimics many gating phenotypes of complex K[+] channels. For their structural simplicity and robust functional properties, these miniature-sized proteins are also the most suitable building blocks for engineering synthetic channels according to the blueprint of complex K[+] channels or with new design principles.

# The pore module

## Evidence for functional stand-alone pore modules

The idea of a modular evolution and function of complex K[+] channels suggests that isolated pore modules exhibit the basic structural and functional hallmarks of K[+] channels. One early argument in favor of such a pore model was related to the inward rectifier class of K[+] channels (Kir) (Jan and Jan, 1994). They are created by two transmembrane domains, which are a structural homolog to the pore module of $K_V$ channels (Nichols and Lee, 2018) but with little intrinsic voltage sensitivity. This overall structural and functional analogy in the pore of $K_V$ and Kir channels (Fig. 1) suggests that the latter represents an ancestral form of modern K[+] channels (Patten et al, 1999; Nelson et al, 1999; Anderson and Greenberg, 2001).

The first experimental tests of the hypothesis of modular evolution and function of K[+] channels, was provided in experiments in which the pore module was separated in a $K_V$-type channel from their VSDs. It was reported that a full-size bacterial $K_V$-type channel from *Listeria monocytogenes* shows the expected strong outward rectification for this type of channel. After separating the pore module from the VSD the former was still functional as a K[+]-selective channel with a residual voltage dependency (Santos et al, 2008). This led the authors to conclude that the pore module has autonomous functions, including gating. A connection between pore and VSD is presumably only augmenting an intrinsic voltage dependency in the pore module (Santos et al, 2006, 2008). In the context of the evolutionary relationship between K[+] channels and Na[+]/Ca[2+] channels (Fig. 1) it is worth mentioning that a similar functional stand-alone pore was also created by genetic separation of bacterial Na[+] channel pores from their VSDs (Shaya et al, 2011). Collectively, these data support the notion that the pore modules of cation channels are structurally robust and autonomous protein units, which can fold independently of their VSD (Shaya et al, 2011). These pores still maintain basic functional properties of the respective channels, including selectivity and rudimentary gating.

Further experiments in which the pore domain and the VSD in a $K_V$ channel were truncated revealed yet another interesting piece in the puzzle of modular evolution. Zhao and Blunck (2016) discovered that the isolated VSD of the Shaker channel generated a distinct cation-selective conductance when truncated after the fourth transmembrane helix S4. Neither the permeation pathway nor the gating mechanism of this ion-conducting, stand-alone VSD are currently fully understood. But since also mutants of Kv

channels (Tombola et al, 2005) as well as voltage-gated proton channels ($H_V$) (Ramsey et al, 2006) can conduct protons and other cations through their VSDs, it was speculated that cations may follow in the truncated Shaker VSD the same pathway. The finding that conductance of the isolated VSD is like the $H_v$ channel sensitive to Zn[2+] can be interpreted as first evidence for a shared conduction pathway (Zhao and Blunck, 2016) and for a close evolutionary relationship between the isolated VSD and $H_v$ channels (DeCoursey, 2013). The available data on the isolated VSD also indirectly suggest that a decoupling of the pore from the VSD alters the conformation of the latter domain, which is evident from the difference in gating currents between the full-length Shaker channel and its truncated VSD (Zhao and Blunck, 2016). These conformational changes may substitute the mutations in the Shake channel, which promote cation conduction through the VSD (Tombola et al, 2005) and explain why the VSD is not serving as a conducting pore in the full-length Shaker channel.

## The viral K[+] channel pore modules are functional channels recapitulating the canonical filter gate and the inner gate of complex K[+] channels

K[+] channel pore modules, which come even closer to the predicted "primordial" pore of K[+] channels, are provided by the Kcv-type channels. These presumably evolutionary, very old K[+] channels from viruses (Thiel et al, 2011, 2013; Hamacher et al, 2012) represent the bona fide pore module of all K[+] channels (Plugge et al, 2000). These small Kcv-type channels (for K[+] channels from **c**hloro**v**iruses) are encoded by large (ca. 190 nm in diameter) dsDNA viruses, which infect freshwater algae. Information on these ancient viruses, on their large genomes, their putative role in evolution and on the role of their K[+] channels in virus infection/replication has been reviewed elsewhere (Van Etten, 2003; Thiel et al, 2010; Thiel et al, 2013; Siotto et al, 2014; Van Etten et al, 2020). Additional reviews provide complementary information on the general occurrence of ion channels in different families of viruses and on their functional significance in viral pathologies (Nieva et al, 2012; Xia et al, 2022; Jalily et al, 2020; Greiner et al, 2016, 2018; Volovik and Batishchev, 2024). At this point it is only important to mention that the viral K[+] channel monomer proteins are, with less than 100 amino acids, indeed minimal in size (Thiel et al, 2011); the 78 amino acids long $Kmpv_{12T}$ channel from the Micromonas pusilla virus 12T, is to our knowledge the smallest functional K[+] channel known to date (Siotto et al, 2014). If we consider the thickness of the membrane bilayer and the number of amino acids required to form two membrane-spanning domains, there are good reasons to believe that the $Kmpv_{12T}$ channel is at or close to the theoretical limit of amino acids, which are required for generating a functional K[+] channel with a 2TMD architecture.

Despite their miniature size and lack of appreciable extracellular and cytosolic domains, the small viral K[+] channels still exhibit the basic structural and many functional hallmarks of all complex K[+] channels (Asrani et al, 2022). On the structural level, this includes the K[+] channel typical architecture with two TMDs and a pore loop with the canonical amino acid sequence TxVG (Y/F/L)G obligatory for the selectivity filter of all K[+] channels (Fig. 3A). It is interesting to mention here that all three combinations of the three amino acids (GYG, GFG, and GLG) in the filter sequence, which are

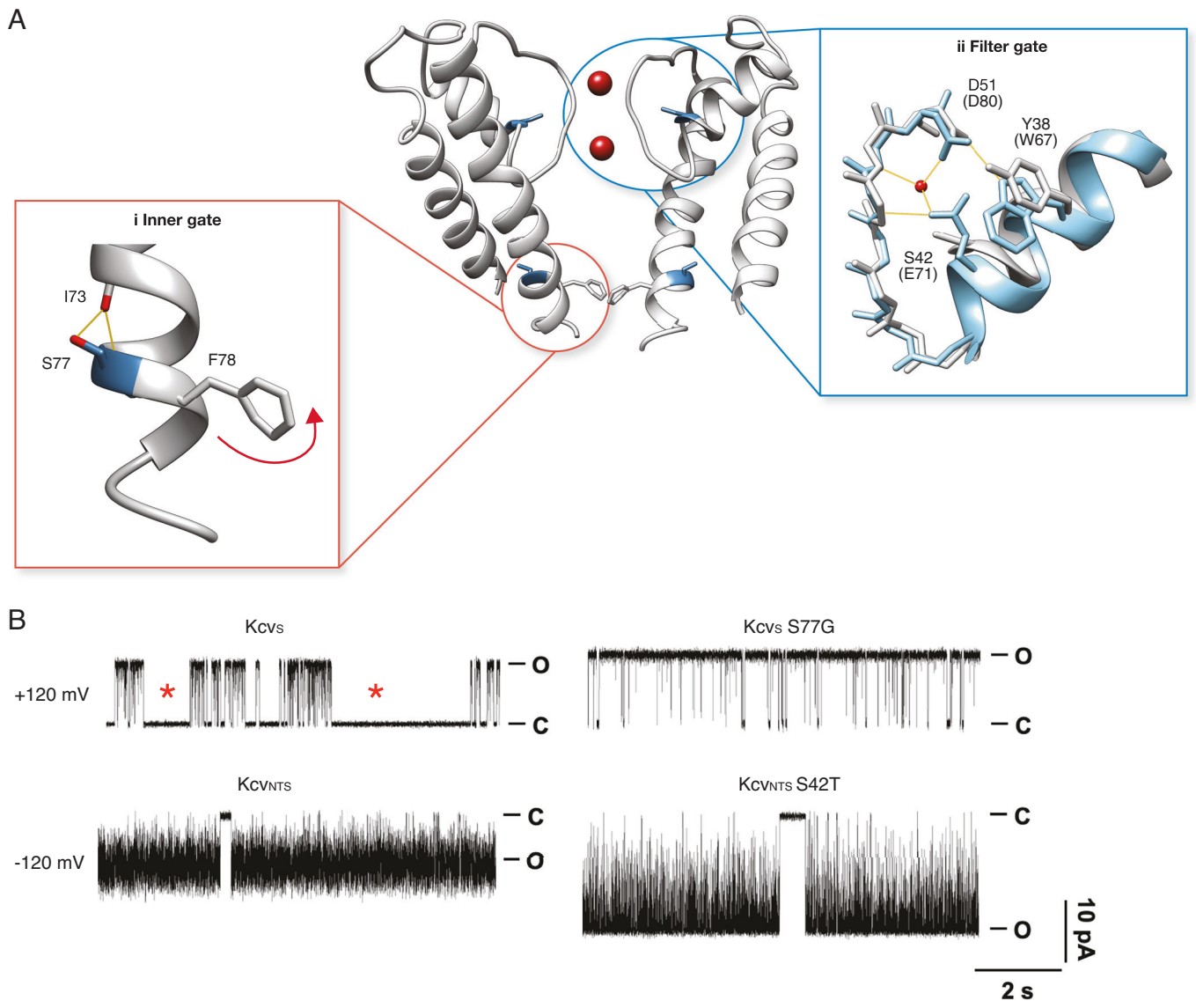

**Figure 3. Location and function of two distinct gates in a model K⁺ channel pore.**

(A) Cartoon representation of Kcv$_{NTS}$ in side-view orientation with two opposing monomers (center) with the position of the inner gate (i) and the selectivity filter gate (ii). Structural details of the two gates are magnified. Instrumental for the inner gate (left) of the Kcv$_S$ channel is a Serine (S77), which can form an inner-helical hydrogen bond with isoleucine (I73). The consequent kink in the helix moves the aromatic Phenylalanine (F78) into the ion-conducting pathway, where it creates a long-lived channel closure. The overall structure of the filter gate is highly conserved in K⁺ channels (right). Structural overlay of Kcv$_{NTS}$ (gray) with KcsA (blue, pbd 1k4c). The critical E71 in KcsA is from its position equivalent to S42 in Kcv channels. Potential hydrogen bonds in KcsA between pore helix and selectivity filter are drawn in orange, including the role of a buried water molecule (Rauh et al, 2021). (B) On the functional level the inner gate (top trace, left) generates in Kcv$_{S-wt}$ a very long-lived closed state (marked by *). Elimination of intrahelical H-bonding and kink formation of the transmembrane helix by S77G mutation eliminates the long-lived closed state; remaining are only fluctuations from filter gating (top trace, right). The filter gate is responsible for a voltage-dependent flicker gating, which is at negative voltages so fast that the full open state is in Kcv$_{NTS-wt}$ not fully resolvable (bottom trace, left). The mutation S42T in the pore helix behind the selectivity filter (A), which anchors the position of the selectivity filter in K⁺ channels (Rauh et al, 2021), stabilizes and slows down the filter gate in Kcv$_{NTS}$ S42T to such an extent that the full open state amplitude of the channel is well resolved (bottom trace, right). Single-channel currents were recorded at indicated voltages with symmetrical 100 mM KCl (10 mM HEPES/KOH pH 7.0) in DPhPC bilayers. C and O highlight the closed and open levels, respectively. Details on channel recordings are given in Rauh et al, 2021, 2022.

present in all complex K⁺ channels, are also found in the library of functional Kcv-type channels (Kang et al, 2004).

This conserved pore module architecture guarantees in the small viral channels the same mechanical connectivity between the two transmembrane domains as in the pore of complex pro- and eukaryotic K⁺ channels (Schmidt et al, 2021). This suggests that the viral channel undergoes the same conformational changes that are

also displayed by the pore units of complex K⁺ channels during gating.

In terms of function viral K⁺ channels resemble those from bacteria and eukaryotes. They have a cation selectivity, which is compatible with all K⁺ channels (Gazzarrini et al, 2003), as they conduct K⁺ and Rb⁺ very well and are impermeable to Li⁺ (Pagliuca et al, 2007). When functionally reconstituted in planar lipid

bilayers, Kcv mediated currents exhibit under bi-ionic conditions with 100 mM $Na^+$ and $K^+$ on opposite sides of the membrane no detectable reversal voltage (Pagliuca et al, 2007) indicating a $Na^+$/$K^+$ selectivity ratios of ~1:100; this value is compatible with the selectivity of complex $K^+$ channels (Hille, 2001). The Kcv channels are, like other $K^+$ channels (e.g., Yohannan et al, 2007) blocked by $Ba^{2+}$ and quaternary ammonium ions in a voltage-dependent manner (Pagliuca et al, 2007; Chatelain et al, 2009; Gabriel et al, 2021; Rauh et al, 2022; Tewes et al, 2024). These functional similarities between Kcv channels and other $K^+$ channels extend even to very detailed features such as a much higher sensitivity of block from the cytosol versus the extracellular side by $Ba^{2+}$ (Tewes et al, 2024). Unlike many eukaryotic $K^+$ channels, the viral $K^+$ channels investigated so far exhibit only a low affinity to a block by tetraethylammonium ($TEA^+$) from the external side. This difference in pharmacology could, however, be abolished by a single point mutation close to the selectivity filter making the Kcv channels very sensitive to $TEA^+$ block (Tan et al, 2012; Winterstein, 2020).

The combination of structural simplicity, robust channel function and functional diversity makes Kcv channels very interesting model systems for understanding the basic structure/function correlates in $K^+$ channels. One interesting aspect of Kcv channels for understanding basic structure/function correlates in $K^+$ channels is the inherent high mutation rate of viruses and their great abundance in the environment, which provides us with a huge library of channel variants. From sequencing endeavors of viruses, which infect 5 different kinds of unicellular green algae from fresh water (e.g., Murry et al, 2020), we have currently a library of ca. 100 different sequences for putative Kcv-type channels. They differ in length between 82 and 98 amino acids but share only 10% sequence identity. If we consider only the well-studied 47 sequences of Kcv$_{ATCV}$-type channels from viruses, which infect the alga *Chlorella heliozoae* as a host, we find that they are all between 82 and 89 amino acids long. Still, they share only a global identity of 30%.

This library of viral $K^+$ channels is further enriched by related but phylogenetically separated sequences of putative $K^+$ channel proteins from viruses, which infect marine algae (Siotto et al, 2014; Kukovetz et al, 2020). At this point it should be mentioned that all of the ca. 30 putative $K^+$ channels from freshwater viruses, which were so far tested, proved to be functional $K^+$ channels (e.g., Kang et al, 2004; Gazzarrini et al, 2006, 2009; Rauh et al, 2017; Murry et al, 2020). Also, some of the candidate proteins from marine viruses were positively tested for $K^+$ channel function. But in this case, not all examined proteins exhibited a detectable channel activity (Siotto et al, 2014; Kukovetz et al, 2020). The reason why some of them, which contain all the essential hallmarks of $K^+$ channels, are not functional, is currently not known.

Extensive functional studies of viral $K^+$ channels on the level of single channels or their macroscopic currents have revealed that the above-mentioned Kcv$_{ATCV}$ channels contain the same two types of gates, namely the selectivity filter gate and the inner gate, which operate in all $K^+$ channels. Figure 3B shows examples of single-channel fluctuations of Kcv channels which were functionally reconstituted in planar lipid bilayers (Rauh et al, 2021). The selectivity filter gate, which is conserved in all Kcv channels, generates a very fast and voltage-dependent closing at negative voltages (Rauh et al, 2022). The finding that this gate is, like filter gating in other channels, sensitive to the concentration of extracellular $K^+$ (Rauh et al, 2017, 2018) and dependent on the nature of amino acids that contact the selectivity filter (Fig. 3Aii, Rauh et al, 2022; Tewes et al, 2024), suggests a common structural and functional mode of operation of the filter gate in all types of $K^+$ channels.

Comparison of two very similar channels yet displaying markedly different open probability (Kcv$_S$ of ca. 0.4 while Kcv$_{NTS}$, >0.9) (Rauh et al, 2017) led to the identification of an inner gate in Kcv$_S$ at the entrance/exit between cavity and the inner solution, not present in Kcv$_{NTS}$. This gate is due to the aromatic side chains of Phe (F78) that move into the ion pathway and close the channel (Fig. 3Ai,ii). A combination of site-directed mutagenesis experiments and molecular dynamics (MD) simulations of a homology structure revealed that the gate is operated by a critical serine (S77) which is absent in Kcv$_{NTS}$. S77 forms an intrahelical hydrogen bond and the resulting kink in the backbone chain moves the aromatic side chain of F78 in the ion-conducting pathway. The presence (Kcv$_S$) or absence of this gate (Kcv$_{NTS}$) in this subgroup of Kcv-type channels (Rauh et al, 2017) provides an excellent experimental system for studying only filter gating or a combination of inner gate and filter gate.

In other types of Kcv channels from virus PBCV1 (Kcv$_{PBCV1}$) a different mechanism of gating at the entrance to the cavity was observed. These channels are slightly larger than Kcv$_{ATCV}$ with a monomer size of 94 AAs and have a short cytosolic helix (12 AAs) in the N-terminus reminiscent of the slide helix in Kir channels (Tayefeh et al, 2007, 2009; Andersson et al, 2018). In these channels, the inner gate is located at the last amino acid Leu94. When the carboxyl group interacts with a $K^+$ ion, this complexed $K^+$ ion closes the inner gate. The channel opens and allows transport when the C-terminus forms a salt bridge with a cationic amino acid (K6) in the N-terminal helix (Tayefeh et al, 2007; Hertel et al, 2010).

Notwithstanding the molecular mechanism of this gate, the data confirm that the Kcv channels have, with respect to their position in the channel protein, a gate which is functionally equivalent to the inner gate of other $K^+$ channels. Like in complex $K^+$ channels, this gate determines the passage of $K^+$ ions between the cavity of the channel and the cytosolic solution.

## Distinct gating phenotypes of complex ion channels can be reconstructed in the primitive isolated pore modules

The modular architecture of $K^+$ channels belonging to the $K_V$ family predicts that peripheral sensor domains perceive physical and/or chemical cues and transmit this information to the gates in the pore domain. In such a scenario, the entire process of gating is occurring in the pore domain. Therefore, we predict that the pore domain of a primitive $K^+$ channel should undergo in a rudimentary manner the same gating movement, induced in complex $K^+$ channels by the sensory domains. This prediction has been met in several studies in which typical gating phenotypes of complex channels were reproduced in simple $K^+$ channel pore models without peripheral sensor domains.

### Slow-activating outward rectification
Outward rectifying $K^+$ channels are closed at negative voltages and open upon depolarization. This distinct type of voltage dependency

is generated, in different manners, by two types of channels, namely $K_v$ and K2P channels. In $K_V$ channels, outward rectification is achieved with the VSD, that mechanically transmits its movement to the gates in the pore. With this distinct voltage dependency and slow kinetic properties of activation and deactivation, outward rectifying $K_V$ channels are instrumental for the repolarization of action potentials in excitable cells.

The family of K2P channels is built according to the 4TM blueprint (Fig. 1) in which each monomer comprises two copies of the pore module; two monomers assemble in the functional tetrameric channel. It has been shown that many members of the K2P channels, which are all activated by diverse chemical and physical signals, also exhibit a distinct outward rectification (Schewe et al, 2016). Despite the absence of a VSD domain, most of the 15 members of the human K2P family exhibit upon depolarization a slow channel activation and a time-dependent deactivation at negative voltages. In this case, voltage sensitivity is an inherent function of the pore module attributed to the selectivity filter gate (Schewe et al, 2016). Outward rectification is an important functional feature of K2P channels in excitable cells where they stabilize the resting potential and contribute to the repolarization of action potentials.

Although $K_V$ and K2P channels generate at first glance a similar phenotype of a time and voltage-dependent outward rectification, they still exhibit distinct differences in the mechanism of rectification. A direct comparison has shown that the voltage dependency of $K_V$ channels occurs at more negative voltages and is considerably steeper than that of any K2P channel (Schewe et al, 2016). A further important difference between the two types of channels is their activation kinetics. While this parameter is in Kv channels sensitive to voltage it is insensitive in K2P channels.

In the context of the modular building principle of $K^+$ channels, it can be concluded that the pore of a $K^+$ channel has an inherent ability to give rise to a time-dependent outward rectification with slow activation and faster deactivation. We can further assume that the evolutionary acquisition of a VSD endowed $K_V$ channels with functional features, which are beyond the simple rectification of the pore only. This includes a negative shift of the voltage window for activation, an increase in the dynamic range of voltage sensing and more capabilities for fine-tuning their kinetics.

Mutational studies on Kcv channels confirmed the assumption that a time-dependent outward rectification is an inherent property of $K^+$ channel pores. Functional reconstitution of the sensor less $Kcv_{NH}$ S77G mutant in planar lipid bilayers shows that this channel functions, like K2P channels, as a slow-activating outward rectifier. Figure 4 show a macroscopic current (A) and the corresponding current/voltage relationship (B) generated from several hundred of $Kcv_{NH}$ S77G channels in a planar lipid bilayer. The channels are mostly closed at negative voltages and open, very much like voltage-dependent K2P channels, slow at positive voltages; a faster deactivation is promoted by membrane hyperpolarization. The mutation S77G removes in this case the inner gate of the channels and amplifies in this manner the properties of its filter gate (Rauh et al, 2017; Winterstein et al, 2021). Hence, the phenomenon of outward rectification of $Kcv_{NH}$ S77G (Fig. 4A,B) must be interpreted as a property of the filter gate just like slow outward rectification in K2P channels. Single-channel data analysis shows that the channel has a high open probability at positive voltages which decreases towards negative voltages (Winterstein et al, 2021).

This decrease in open probability results from very long closed times which appear with increasing frequency at membrane hyperpolarization. The open probability of the channel can be fitted with a Boltzmann function yielding a half maximal activation at about $-60$ mV and a gating charge value of ca. 0.8 (Winterstein et al, 2021).

These data cannot be directly compared to recordings of K2P channel activity; the latter were mostly performed in cells and under asymmetrical conditions with $Rb^+$ in the cytosolic buffer, e.g., factors which affect channel gating (Schewe et al, 2016). With these cautions in mind, the data still indicate that the outward rectifying Kcv mutant opens at a similar range of voltages but with a lower voltage sensitivity than K2P channels (Schewe et al, 2016).

Collectively, the data once again indicate that slow outward rectification is an inherent property of a $K^+$ channel pore module and is mediated by filter gating. Finding this phenotype of gating in a Kcv channel also means that it does not require any specific structural features of K2P channels, like an asymmetry in the selectivity filter.

### $K_V$ type inward rectification

Some $K_V$ family members respond to voltage in the opposite way. They are closed at depolarized voltages and open in a slow manner upon hyperpolarization (González et al, 2012). These so-called inward rectifying channels like KAT1, are found in plants (Thiel and Wolf, 1997), where they catalyze the influx of $K^+$ ions at voltages negative of the $K^+$ equilibrium voltage into cells. In mammalian cells the so-called Hyperpolarization-activated cyclic nucleotide-gated (HCN) channels exhibit a similar voltage-dependent activation (Wahl-Schott and Biel, 2009). Because of the weak cation selectivity of HCN channels, they catalyze at negative voltages an influx of $Na^+$ into the cells, which in turn causes a membrane depolarization and eventually the triggering of an action potential. HCN channels have a crucial physiological function in the control pacemaking of the heartbeat and of the firing frequency of neurons.

The mechanistic difference between outward and inward rectification of $K_V$ type channels is still not fully understood. Comparative analyses reveal that inward and outward rectifiers share the same overall architecture, including in both cases S4 elements with positive net charges. It was even shown that the movement of the critical S4 segment, which serves as the main sensor for changes in the electrical field, is similar in channels, which are activated by depolarization or hyperpolarization (Männikkö et al, 2002). Recent studies suggest that in the case of HCN channels, the direction of rectification seems to be determined by interactions between the VSDs and the pore domain at the level of the inner gate (Ramentol et al, 2020).

In the context of this hypothesis, it is interesting that the gating phenotype of a slow-activating inward rectifying $K_V$ channel can be mimicked by mutations at the level of the inner gate in a Kcv channel, even though the latter lacks a voltage sensor domain. When the wt $Kcv_{PBCV1}$ channel is expressed in mammalian cells it exhibits in whole-cell patch-clamp recordings an instantaneous current largely voltage-independent, with a small degree of inward rectification (Moroni et al, 2002; Hertel et al, 2006; Baumeister et al, 2017). Mutations introduced in the outer transmembrane domain (Fig. 4C) (P13A) (Hertel et al, 2006, 2010) or extension of the same TM domain (Baumeister et al, 2017), reproduce the time-

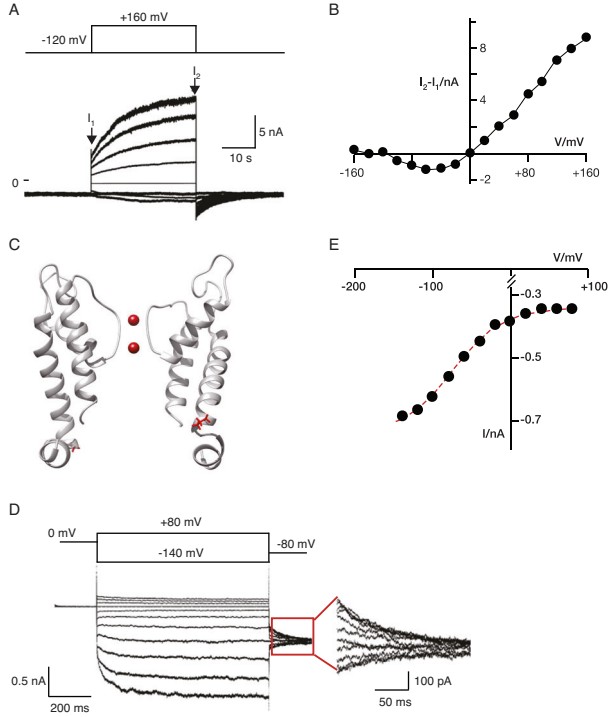

**Figure 4. Slow outward and inward rectification of Kcv channels.**

Outward rectification: (A) Multi-channel current response of slow outward rectifying $Kcv_{NH}$ S77G channels in a DPhPC bilayer. Currents were evoked in symmetrical 100 mM KCl by stepping voltage from a holding value of −120 mV for 30 s to test voltages between −120 and +160 mV in 40 mV steps before stepping voltage back to −120 mV. The zero-current level is indicated by a black bar. Instantaneous currents and steady-state currents are marked by $I_1$ and $I_2$, respectively. (B) I/V relationship of time-dependent current ($I_2$–$I_1$) (for further detail see Rauh et al, 2021). Inward rectification: (C) Cartoon representation of $Kcv_{PBCV1}$ monomer in which one Ala (red sticks) was inserted in position 17 at the beginning of the first transmembrane helix. (D) Slow-activating inward currents of this mutant were elicited in HEK293 cells by voltage steps from +80 to −140 mV in 20 mV increments during which the currents exhibited a slow activation. After reaching steady state, the membrane was stepped from the conditioning voltages to −80 mV to follow the relaxation of tail currents (tail currents from red box magnified on the right). Because the tail currents experience at −80 mV the same driving force their difference is entirely determined by the channel open probability prior to the step to −80 mV. (E) Plot of initial tail currents as a function of the conditioning voltage provides measure for the voltage dependency of slow- activating current. The data were fitted by a Boltzmann function yielding a value of −75 mV for $V_{1/2}$ and a value of 0.83 for z. Data in (E, D) reproduced with permission from Baumeister et al, 2017. Voltage protocols for measuring currents in (A, D) are shown above the current traces.

dependent slow-activating inward rectification of KAT1 and HCN channels (Fig. 4D,E) superimposed on an instantaneous current. Like in canonical inward rectifying $K_V$ channels, current activation kinetics are in the Kcv enhanced by negative voltages; at positive voltages, the currents decrease in a time and voltage-dependent manner (Baumeister et al, 2017). An analysis of the voltage dependency of the slow-activating inward current from tail currents, shows a $V_{1/2}$ value at about -80 mV and an average gating charge (z) of about 0.8 (Fig. 4E). The latter value gives the valency of the charge which is moved in the electrical field during channel gating; it provides an indirect estimate of the voltage

dependency of a channel. This means in the case of the Kcv channel that the responsible 'voltage sensor' in the pore module senses with one elementary charge nearly the entire electrical field. This means that the channel has a low voltage dependency. But also, some canonical $K_V$ channels can exhibit rather low degrees of voltage dependency. The $K_V$-type inward rectifier KAT1, for example, has a z value of 1.4 (Hertel et al, 2005) meaning that the presence of an entire VSD including a S4 domain only augments in this channel the voltage sensitivity only by a factor of two over that in the stand-alone pore module.

Molecular dynamic simulations of $Kcv_{PBCV1}$ (Tayefeh et al, 2009) as well as experimental measurements of the temperature sensitivity of channel opening, suggest that in the inward rectifying mutants, the voltage-dependent step is the formation and breaking of the critical salt bridges of the inner gate (Baumeister et al, 2017). Considering the general hypothesis that the voltage-sensing domain is only operating gates, which are already existing in the channels, we propose that the $K_V$ inward rectifiers function by opening the inner gate in a manner that is mimicked by the Kcv pore mutants. This speculation is realistic when we consider that a tilting of the cytosolic part of S5 and S6 was indeed observed in the transition between closed and open pore structures (Saponaro et al, 2021) as well as in coarse-grained computational simulations of HCN channels (Gross et al, 2018). In this context, it is also interesting to mention that in HCN1 a mutation (G391D) in S6 at the entrance to the cavity abolishes the activity of this channel (Marini et al, 2018). From MD simulations, we learn that the mutant mimics with its anionic amino acid the aforementioned mechanism at the inner gate of $Kcv_{PBCV1}$ in that the negative charges at 391 are trapping a $K^+$ ion at the inner gate. As a result, permeation is prevented in this mutant (Marini et al, 2018).

### Kir-type inward rectification

Inwardly rectifying potassium (Kir) channels are, in the superfamily of the modern $K^+$ channels, structurally most similar to the primordial pore module of all $K^+$ channels (Fig. 1). In addition to the membrane-embedded pore though, they contain large and functionally important cytosolic domains. Kir channels exhibit by themselves no apparent intrinsic gating (Nichols and Lee, 2018). Their voltage-dependent inward rectification is generated by a distinct channel blockade mediated by endogenous polyamines and $Mg^{2+}$ (Hibino et al, 2010). Both divalent and polyvalent cations enter the channel pore with positive voltage from the cytosolic side in a voltage-dependent manner to reach their specific binding sites in the pore domain. Upon binding to these specific sites, they block the conductance of $K^+$ ions (Baronas and Kurata, 2014). This block is then released by negative voltages, where a $K^+$ inward current expels the blocking polyamines and $Mg^{2+}$ from their pore binding sites.

In a screening of viral $K^+$ channels from environmental sea water samples, we detected a small protein ($Kmpv_{SP1}$) of 87 amino acids which exhibits in mammalian cells or in planar lipid bilayers the functional hallmarks of Kir-type channels (Eckert et al, 2019). Figure 5 shows the structure of $Kmpv_{SP1}$ predicted by AlphaFold (Fig. 5A) and the inward rectifying conductance measured by whole-cell patch-clamp recordings in HEK293 cells expressing the protein (Fig. 5B,C). In a Kir channel type manner, the viral protein exhibits only a small outward current at voltages positive of the $K^+$ equilibrium voltage and a large conductance at more negative voltages. Most importantly, like in Kir channels the voltage

dependency of gating shifts also in the viral protein with the $K^+$ reversal voltage. Like mammalian Kir channels, the viral proteins are also highly $K^+$ selective and blocked in a voltage-dependent manner by $Ba^{2+}$ (Eckert et al, 2019). But despite all these similarities the mechanism underlying inward rectification must be fundamentally different from the voltage-dependent block in Kir channels. In the case of the viral channels, the inward rectification is an inherent property of the channel pore. Notably, rectification on the level of the single channel also occurs when the protein is functionally reconstituted in planar lipid bilayers in KCl-only buffer (Fig. 5D,E). Rectification was maintained even after the addition of 1 mM EDTA to eliminate any potential contamination of $Mg^{2+}$ (Eckert et al, 2019). Also, different from Kir channels, the viral protein does not require the cytosolic domains (Eckert et al, 2019), which are very important determinants of rectification in Kir channels (Baronas and Kurata, 2014).

In conclusion, the available data indicate that the isolated pore module of a $K^+$ channel can have intrinsic gating properties, which resemble those of modern Kir channels. This finding agrees with some isolated reports, which have already shown that distinct mutations were able to induce some intrinsic inward rectification in Kir channels independent of blockage by divalent cations (Lu and MacKinnon, 1994; Chang et al, 2007). An extensive study of mutants and a comparative analysis of the inward rectifying $Kmpv_{SP1}$ channel with a similar non-rectifying viral channel provided no clear-cut mechanistic explanation for the rectification of the former (Eckert et al, 2019). These experiments so far can only exclude a peculiarity in the selectivity filter of $Kmpv_{SP1}$ as source of intrinsic rectification. The data also exclude single amino acids as crucial for rectification. It seems more likely that the folding of the transmembrane domains plays a key role in this type of gating.

## Complex $K^+$ channels with their respective functional properties can be engineered by fusion of Kcv channels with peripheral sensory domains from evolutionary unrelated sources

The preceding chapter has shown that the isolated pore module of $K^+$ channels can generate distinct phenotypes of gating that are otherwise found in complex channels by adding sensory components. In these channels, peripheral sensor domains perceive environmental cues and transmit them either to the inner gate or the filter gate in the pore domain. Such a concept has been beautifully demonstrated already for K2P channels (Bagriantsev et al, 2011), in which protons, heat, and pressure modulate the activity of these channels via a common gate, presumably the filter gate. In the following chapters, we review the experimental support for peripheral modulation of pore gating, in the case of voltage-gated and ligand-gated channels.

### Outward rectifying $K_V$ channel
$K_V$ channels were presumably formed in evolution by fusion of genes which code for the regulatory modules to the channel pore. In this way the pore could acquire its typical regulation by physical (voltage) or chemical (e.g., ligands) stimuli.

This concept of modular evolution was supported by experiments in which portions of the S1–S4 domain, which form the VSD in the eukaryotic $K_V2.1$ channel (Fig. 2i), were substituted by equivalent domains from other proteins (Alabi et al, 2007; Bosmans

et al, 2008). The latter include the evolutionary distant archae-bacterial channel $K_VAP$, the proton-conducting voltage-sensing $H_V1$ (Fig. 2ii) and the voltage-sensitive sodium channel $Na_V1.2a$. The finding that chimeras with paddle domains from different donor proteins in a eukaryotic $K^+$ channel were generating functional channels with distinct voltage sensitivity strongly support the idea of an evolutionary patchwork. Functional characterization of chimeras stresses that the "paddle" motif is a central unit for mediating voltage-dependent gating in different proteins. Each of the tested VSDs has its characteristic voltage dependency which is reflected in the gating in the resulting chimera. Additional data on the toxin sensitivity furthermore underline that the paddle motive does not only confer its voltage dependency on the chimera but carries also a distinct toxin sensitivity to the composite channel protein (Alabi et al, 2007; Bosmans et al, 2008).

One of the milestones in support of a modular evolution and function of $K_V$ channels was the discovery of voltage-sensitive phosphates (Ci-VSP) in the tunicate *Ciona intestinalis* (Fig. 2iii). These proteins are built from a membrane resident domain, Ci-VSD, which resembles the VSD of $K_V$ channels, and a cytosolic phosphatase (Murata et al, 2005). Because of the positive net charge in the Ci-VSD the protein undergoes upon membrane depolarization a minute movement in the bilayer perpendicular to the membrane plane. This small charge displacement can be measured and is termed "gating current". A comparison of gating currents from the Ci-VSP protein and from $K_V$ channels in response to membrane depolarization shows that they are similar (Murata et al, 2005; Okamura et al, 2009). And like in the case of ion channels, this motion of the voltage sensor domain is transmitted into a control of the cytosolic activity of the phosphatase in hydrolyzing phosphoinositides. The modular control of the voltage sensor over a cytosolic protein has been nicely shown in a protein engineering endeavor in which the Ci-VSP was linked to the tumor suppressor PTEN. Functional testing of this synthetic protein confirmed that a functional coupling of two proteins via a suitable linker domain generated a PTEN-like enzymatic activity under the control of the VSD from Ci-VSP (Lacroix et al, 2011).

Another demonstration of a potential modular evolution of $K^+$ channels came from the successful fusion of different building blocks to form functional $K_V$ channels. In many cases, it was possible to swap the pore domains and the VSDs between different $K^+$ channels with the result that the synthetic channel acquired the voltage dependency of the protein donating the VSD (Patten et al, 1999; Alabi et al, 2007). While these swapping experiments were successful within the same family of $K^+$ channels, they largely failed when domains from unrelated proteins were fused. For example, the pore module of the KcsA channel was joined with the VSD of canonical $K_V$ channels. In these studies, the naïve coupling of VSD of a $K_V$ channel and the pore domain of KcsA did not generate a measurable current (Caprini et al, 2001); only small parts of the KcsA pore could be transferred into the scaffold of a $K_V$ channel without compromising function (Lu et al, 2001). To obtain a functional channel, parts of the $K_V$ pore module with the N-terminal end of S5 and the C-terminal end of S6 had to be included in the chimera. These data suggest that orthogonal domains of modern channels may be structurally incompatible and that a distinct coupling between these modular domains is required for functional interaction (Caprini et al, 2001). These findings

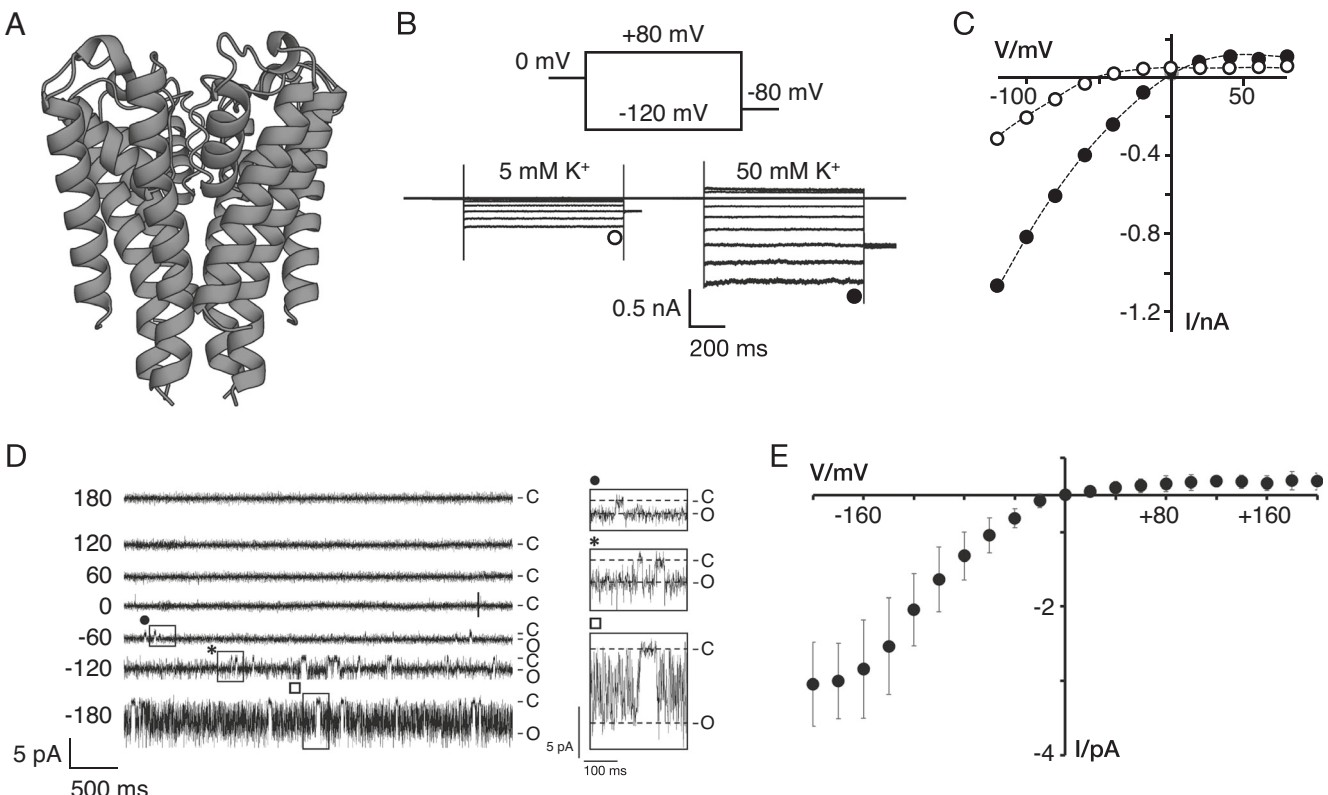

**Figure 5. KmpvSP1 generates inward rectifying conductance.**

(A) AlphaFold predicted structure of KmpvSP1 channel; each monomer in the tetramer is made of 87 amino acids. (B) Expression of KmpvSP1 in HEK293 cells elicits an inward rectifying current in which rectification shifts with K+ driving force. Typical current responses in a HEK293 cell expressing KmpvSP1 to voltage steps between +80 and −120 mV with 130 mM K+ in the cytosolic buffer and with 5 mM (open symbol) or 50 mM (solid symbol) K+ on the extracellular side (C) Steady-state I/V relation of data from A shows large inward currents only negative of K+ reversal voltages for 5 and 50 mM K+. (D) Functional reconstitution of KmpvSP1 in planar lipid bilayers with symmetrical 100 mM KCl generates channel activity only at negative voltages. At these voltages, the channel is mostly open and only interrupted by short closures. The boxed areas in the current traces are highlighted at higher resolution on the right. (E) Mean channel currents sampled over one minute at respective test voltages. For experimental details, see Eckert et al, 2019.

questioned the hypothesis of a one-step fusion between domains as the basis of the evolution of $K_V$ channels.

The ultimate test of the modular composition of $K_V$ channels eventually came when the entire VSD from Ci-VSP of *C. intestinalis* was fused to a Kcv channel to generate the first functional synthetic $K_V$ channel $Kv_{Synth1}$ (Fig. 6A) (Arrigoni et al, 2013). Currents recorded from *Xenopus laevis* oocytes expressing $Kv_{Synth1}$ show that the naïve connection of the two elements transforms the voltage-independent Kcv channel into a canonical $K_V$-type outward rectifier with slow-activating kinetics (Fig. 6A) (Arrigoni et al, 2013). This study clearly shows for the first time that it is indeed possible to engineer complex synthetic channels in which the pore gating is under the control of a sensing domain. Worth mentioning in this context is also the high degree of structural plasticity of the two modules. Notably, the Ci-VSD functions in its native protein as a monomer (Okamura et al, 2009) while it imposes in the case of $Kv_{Synth1}$ voltage dependency to a tetramer.

A scrutiny of the functional properties of $Kv_{Synth1}$ shows that the synthetic channel exhibits the features of both elements, namely the channel pore and the voltage sensor (Arrigoni et al, 2013): the

permeability properties of Kcv are preserved in $Kv_{Synth1}$. The channel discriminates between K+ and Na+ ($P_{Na}/P_K$ of 0.04) as much as Kcv. Also, $Kv_{Synth1}$ maintains the general pharmacology of the Kcv pore. Analysis of the activation curve of the slow and voltage-dependent outward current in $Kv_{Synth1}$ shows, on the other hand that the channel is activated by depolarizing voltages with an apparent $V_{1/2}$ of +56 mV and requires the movement of one electric charge across the membrane ($z = 0.92$). Half activation voltage and equivalent charge values of $Kv_{Synth1}$ resemble those reported for Ci-VSP; hence after coupling the voltage sensor to the pore the former imposes its voltage-sensitive conformational changes onto the pore gates. This idea is further supported by the finding that a double mutation R229Q/R232Q in the S4 segment of the VSD, which eliminates the gating current of Ci-VSD and generates a voltage-insensitive phosphatase, also eliminates voltage sensitivity in $Kv_{Synth1}$ (Arrigoni et al, 2013). Since the pore in $Kv_{Synth1}$ maintains the filter gate properties of the isolated pore it is reasonable to speculate that the movement of the Ci-VSD causes a modulation of the inner gate.

A recurrent problem in the design of synthetic proteins is the question on how to link two orthogonal proteins for transmitting

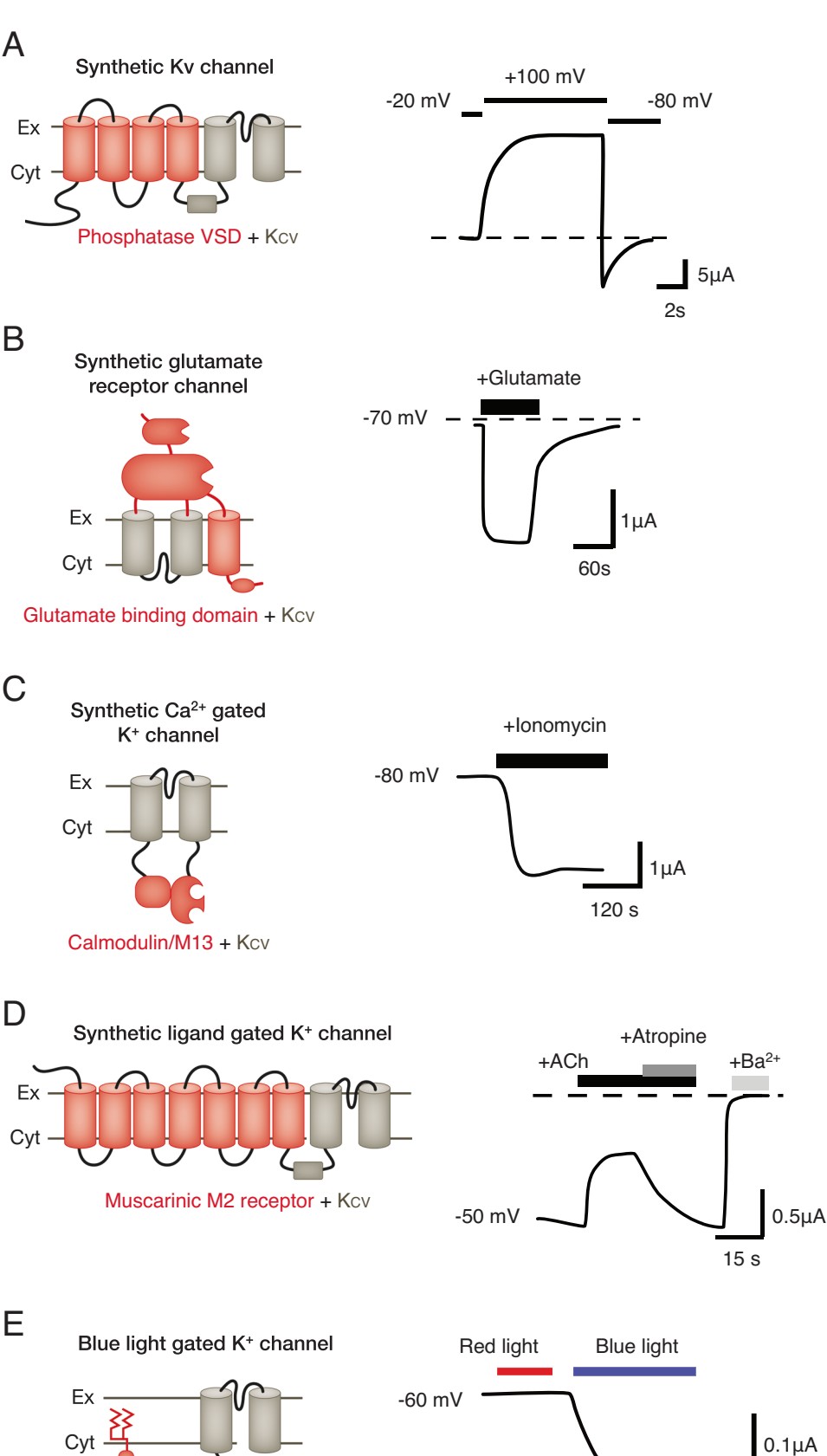

© The Author(s)

**Figure 6.  Kcv channels as modular components of synthetic proteins.**

Cartoon representation of different synthetic channels based on fusing regulatory domains to Kcv channels (left) and gain of functional features in these channels (right). The respective current traces measured at indicated voltages are schematically redrawn from published data; dotted lines indicate zero current. The common Kcv channel scaffold is shown in gray, the additional domains in red. The extracellular and cytosolic sides of the membrane are indicated by Ex and Cyt, respectively. (**A**) Fusion of $Kcv_{PBCV1}$ to voltage-sensing domain Ci-VSD from a *Ciona intestinalis* phosphatase. The construct $Kv_{Synth1}$ generates in *Xenopus* oocytes a slow-activating outward rectifier (Arrigoni et al, 2013). (**B**) After replacing the transmembrane domains of the mammalian AMPA GluA1 glutamate receptor with $Kcv_{ATCV1}$ the synthetic channel inserts into the plasma membrane of *Xenopus* oocytes in an inverse orientation. Addition of 100 µM Glutamate to the external medium causes a rapid and reversible activating $K^+$ current (Schönrock et al, 2019). (**C**) $Kcv_{PBCV1}$ with calmodulin attached to the C-terminus and the calmodulin binding protein M13 to the N-terminus. Elevation of cytosolic $Ca^{2+}$ concentration following treatment with 1 µM ionomycin generates an increase in channel conductance (DiFrancesco et al, 2015). (**D**) N-terminal fusion of human muscarinic M2 (hM2) receptor to $Kcv_{PBCV1}$. Application of 5 µM Acetylcholine (ACh) inhibits current while the antagonist Atropine at 1 µM inverses this effect leaving a remaining current, which is blocked by $Ba^{2+}$ (García-Fernández et al, 2021). (**E**) N-terminal fusion of $Kcv_{PBCV1}$ to a construct comprising a farnesylation domain for membrane anchoring and parts of a <u>L</u>ight-<u>O</u>xygen-<u>V</u>oltage (LOV) sensing domain from *Avena sativa* plus J-α domain for light sensing. This synthetic channel is activated by 50 µW/mm² blue light but not by 50 µW/mm² red light (Cosentino et al, 2015).

conformational information between them. In the construction of $Kv_{synth1}$ this problem was tackled by trial and error. In a first attempt the VSD was directly connected to the Kcv channel without the linker sequence that connects the Ci-VSD to the phosphatase in *C. intestinalis*. In this case the small N-terminal helix of Kcv, which can be seen in Fig. 6A, acts as a linker. To determine the general influence of the linker on gating also other constructs with different portions of the N-terminal helix of Kcv and the Ci-VSD/ phosphatase linker were tested. All constructs produced functional outward rectifier channels in *Xenopus laevis* oocytes (Arrigoni et al, 2013), suggesting that the precise linker sequence is not important for the functional coupling. Interesting to mention, however, is that the degree of rectification was negatively correlated to the length of the linker. Hence, it seems as if the mechanical transmission depends in this synthetic channel more on length than fold of the linker. These findings agree with the concept of a rigid coupling between sensor and pore in $K_v$ channels (Blunck and Batulan, 2012) with the aforementioned "swapped" architecture.

The dependency of rectification on the length of the linker, which connects the VSD from Ci-VSP and Kcv channel pore suggest an important contribution of this linker for mediating the conformational changes in the voltage sensor to the gate in the channel pore.

Collectively, the results of this study confirm the hypothesis that a naïve coupling of orthogonal protein domains could have occurred during evolution for generating the precursor of modern $K_V$ channels (Nelson et al, 1999; Anderson and Greenberg, 2001). This primitive channel with its basic voltage dependency may have later evolved with respect to new physiological demands, such as shifting the window of voltage dependency more into the range of physiologically relevant voltages. With the evolution of fast signal transitions in the developing neuronal system also the steepness of the voltage dependency and the speed of activation and deactivation could have been adapted to the physiological needs. The simple experiments on the importance of the length of the S4/S5 linker on the degree on rectification in $Kv_{Synth1}$ (Arrigoni et al, 2013) are already an indication on how the mechanical interplay between the two domains could have been the target of evolution.

### Synthetic glutamate receptor channel

Ionotropic glutamate receptors (iGluRs) and $K^+$ channels have at first glance little in common (Kuner et al, 2003). While the former conduct cations with a low degree of selectivity, the latter can efficiently discriminate between $K^+$ and $Na^+$ ions. Apart from them

sharing the typical features of membrane proteins with membrane-spanning α-helices, their respective 3TMD and 2TMD composition have no striking sequence similarity. However, in the mid 1990's it was realized that the M2 segment of iGluRs and the pore loop of $K^+$ channels have some critical sequence similarity; this led to the proposal that these sequences might form in both type of proteins the ion-conducting pathway (Wo and Oswald, 1995; Wood et al, 1995). The similarity between iGluRs and $K^+$ channels and their possible evolutionary relationship was further supported by the discovery of primitive precursors of modern glutamate receptors. One of these, GluR0 is a bacterial glutamate-activated and $K^+$ selective channel from *Synechocystis* with only two transmembrane domains (Chen et al, 1999). Its $K^+$ selectivity is achieved by the presence of a $K^+$ channel typical pore loop, including the TXVGYG signature sequence of the $K^+$ channel selectivity filter. An evolutionary intermediate between the ancestorial bacterial and vertebrate glutamate receptors is presumably presented by AvGluR1 from the rotifer *Adineta vaga* (Janovjak et al, 2011). While this protein has the same architecture of eukaryotic iGluRs, it still contains a pore loop with the signature sequence of the $K^+$ channel selectivity filter.

These findings have fostered the speculation that modern glutamate receptors may have arisen during evolution from the fusion of primitive $K^+$ channel pores and glutamate-binding sensor domains (Wood et al, 1995; Kuner et al, 2003; Wo and Oswald, 1995). Because experimental attempts to create such synthetic glutamate-regulated channels from $K^+$ channel pores and sensor domains have failed, the hypothesis was subsequently rejected (Hoffmann et al, 2006; Sobolevsky et al, 2003).

Motivated by the experiments showing that small viral $K^+$ channels are very well suited for building synthetic $K_V$ channels with novel gating properties, we revisited the question of whether it is possible to construct a glutamate-regulated channel from independent components. This was done by replacing the transmembrane domain of the mammalian AMPA GluA1 gluta-mate receptor with a simple viral $K^+$ channel $Kcv_{ATCV}$ (Fig. 6B) (Schönrock et al, 2019). The experimental data demonstrate that the resulting construct, $Glu_{ATCV}$, functions in *Xenopus* oocytes as a glutamate-regulated, potassium-selective ion channel. After several design improvements, the synthetic channel exhibited a sensitivity to glutamate (Fig. 6B) like that of the native human iGluRs. These experiments proved an important hypothesis concerning the evolution of glutamate receptors. This makes it very likely that the modern glutamate receptors may have, as proposed by many in

the field, evolved from the fusion of simple $K^+$ channel pores and independent sensing elements.

In addition to advancing our understanding of the evolution of membrane proteins, de novo building of a synthetic channel provides several interesting insights into structure/function correlates in native glutamate receptors. For example, the data showed that the inverse orientation of the small channel pore in the plasma membrane can be controlled by fusion of the channel protein to a signal peptide (Schönrock et al, 2019). By systematically minimizing the structure of the synthetic glutamate receptor, it was possible to address other questions on structure/function details in these proteins. For example, by deleting individual domains, it became apparent that the amino-terminal domains (NTDs) are not as important for iGluR assembly as anticipated. The finding that the synthetic protein also worked well as a glutamate-sensitive channel when M4 was deleted, further implies that these domains are not essential for assembling a minimal functional glutamate-gated channel.

# Channels with new functional properties, which are not present in nature, can be engineered by combining a pore module with orthogonal sensing domains

The examples presented so far have supported the potential of modifying the gating of Kcv channels by protein engineering. They have also established that fusion of orthogonal sensor domains with these simple $K^+$ channel pores lead to channels, which are structurally and functionally similar to proteins that already exist in nature. The results of these experiments also lend support to the general idea on the modular building principle according to which different domains were joined during evolution to the central $K^+$ channel pore domain. In this way, modern channels have presumably acquired a more precise and wider range of control over the gates, which are already present in the pore module. The ability to build synthetic channels according to the blueprint of existing channels prompted the idea of engineering channels with completely new functional properties. Such custom-designed channels in which highly sensitive and selective biological sensor domains can gate an ion channel protein may in the future serve as integrated components in electronic nanodevices for detecting molecules of interest like environmental toxins (Shim and Gu, 2007). Like in the case of optogenetic applications (Fenno et al, 2011), synthetic channel proteins in which the gating is modulated by external cues like as temperature, ultrasound or magnetic field changes may even find future medical applications for a remote modulation of cell excitability. Also, in this endeavor of engineering channels with new functional properties the small viral $K^+$ channels proved to be very suitable building components.

A first proof-of-concept study tried to craft a $Ca^{2+}$ dependency into the $Kcv_{PBCV1}$ channel; this channel is otherwise not sensitive to cytosolic $Ca^{2+}$. The design of the channel was inspired by the $Ca^{2+}$ sensor Cameleon, in which binding of $Ca^{2+}$ causes a conformational change in calmodulin and promotes its binding to M13 peptide (Miyawaki et al, 1997). The experimental data show that a chimera in which a $Kcv_{PBCV1}$ subunit was fused to calmodulin and its interacting peptide M13 (Fig. 6C) exhibited $Ca^{2+}$ sensitive gating (DiFrancesco et al, 2015). When the Kcv/Calmodulin/M13

construct was expressed in *Xenopus* oocytes its conductance increased in response to elevated cytosolic $Ca^{2+}$. The results of these experiments confirmed that orthogonal protein components can be fused to the channel pore in such a way that the basic channel pore gains some rudimentary ligand-sensitive gating.

The modular architecture of channel proteins has also inspired the coupling between ion channels and receptor proteins. These so-called ICCR, for Ion channel-coupled receptor, rely on the fusion of a G-protein-coupled receptor with its extracellular ligand binding domains and an ion channel. With a mutual interplay of both components, it should be possible to combine the selective and specific sensing capacity of the GPCRs with the conductive properties of the channel. This should result in a nanoscale electrical biosensor for extracellular ligands. In a series of studies, GPCRs with distinct ligand binding specificities were fused to different ion channels like Kir6.2 and TREK-1 and tested in cells or cell-free devices for function (Moreau et al, 2008; García-Fernández et al, 2021; Lim et al, 2015; Niescierowicz et al, 2014). The basic message from the experimental characterization of these different ICCR constructs is that the GPCR-specific binding of extracellular ligands is, as expected, causing a measurable change in the conductance of the attached ion channel. This impact on channel gating must be transmitted from the receptor unit by long-range mechanical interactions independent of the G-protein connected to the receptor unit. Interesting for the focus of the present review is that the engineering of ICCRs was also possible with one of the small Kcv channels. In this case (Fig. 6D) it required the optimizing of the linker connecting the channel and the GPCR (in this case the human muscarinic M2 receptor). The result was at least in one case, the conversion of Kcv into a ligand-gated channel. The latter is inhibited by acetylcholine and stimulated by its antagonist atropine (Fig. 6D). While these studies again established that the conformational change of a sensor domain can be used to alter the gates in $K^+$ channels including the simple Kcv channel pore, it also shows that a naïve coupling between domains is not a simple plug and go endeavor. The authors report, for example, that fusion of the same Kcv pore to the dopamine receptor did not result in a functional ligand-gated channel (García-Fernández et al, 2021). These results show that the increasing knowledge on the atomistic structure of channels and sensor domains has provided valuable insights into mechanical coupling in complex proteins. But this information is not yet enough for plug and go design of synthetic channel proteins.

The remote manipulation of ion channels with light, namely optogenetics, provides a powerful technique for controlling cellular activity including neuronal firing with high spatial and temporal precision. Motivated by the goal to create light-gated $K^+$ channels, we and others (Cosentino et al, 2015; Sierra et al, 2018; Janovjak et al, 2011; Fortin et al, 2011; Banghart et al, 2004; Kang et al, 2013) have attempted to engineer synthetic light-gated $K^+$ channels. In pioneering work, we have built a chimeric protein, BLINK1, composed by the plant photoreceptor LOV2 domain and the viral channel $Kcv_{PBCV1}$ (Fig. 6E). After several rounds of optimization, which again was mostly focused on finding the best length for the linker between LOV2 and Kcv, the BLINK1 channel exhibited a reversible blue light-dependent gating, resulting in large $K^+$ selective currents, imposing hyperpolarization of the membrane resting potentials. The increase in $K^+$ conductance was activated by blue light of low intensity in the range $\mu W/cm^2$ (Fig. 6E) and

**Box 1  In need of answers**

- What is the evidence for a modular architecture and a mosaic type of evolution of ion channels?

- What is the minimal structure of the pore module in K$^+$ channels, and does it function by itself?

- Can we find experimental evidence for a modular evolution of complex ion channels by rebuilding proteins via naïve fusion of individual modules?

- Is it possible to build synthetic channels with new functional properties by fusing a pore module with orthogonal signaling domains?

reversed in the dark. Altogether, this system was a versatile fully genetically encoded blue light-sensitive system for inhibiting neuronal activity. Proof-of-concept experiments for an in vivo application, in zebrafish, rats and *A. thaliana* indicated that BLINK1 could indeed be employed as an effective inhibitory tool not only in animals (Alberio et al, 2018) but also in plants (Papanatsiou et al, 2019).

## Summary

In the spirit of the Richard Feynman quote, "What I cannot create, I do not understand" the present review shows that it is possible to recreate the complex gating mechanisms of modern K$^+$ channels in primitive Kcv channels. This is achieved either by mutations in the channel pore or by naïve fusion of the pore to orthogonal sensory domains. From this building endeavors, we understand that a primitive pore domain already contains gates, which can generate complex gating phenomena like voltage- and time-dependent inward or outward rectification. We further understand that conformational changes in attached sensor domains are transmitted to the pore where they affect the function of the inherent pore gates. In this way, it is possible to recreate without the need for coevolution, a synthetic K$_v$-type channel by fusing a Kcv channel to an isolated VSD. Even more remarkable is the finding that a glutamate-gated channel can be created by fusion of the Kcv channel with a glutamate-binding domain. What we understand from these experiments is that modern glutamate receptor channels could have evolved from the fusion of a glutamate-binding domain with a primitive K$^+$ channel pore module. The presentation of some synthetic channels with the Kcv pore as conducting module furthermore suggests that this principle of modular building is suitable as a blueprint for the creation of channel proteins with new functions (Box 1).

## Peer review information

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

## Acknowledgements

The work was funded by Deutsche Forschungsgemeinschaft (DFG, German Research Foundation) project ID 528562393-FIP 26 and grant TH558/34-1 and ERC-2023-SyG n. 101118744 to AM.

## Author contributions

**Oliver Rauh**: Data curation; Formal analysis; Investigation; Methodology, Writing—review and editing. **Tobias Schulze**: Formal analysis. **James L Van Etten**: Conceptualization; Data curation. **Gerhard Thiel**: Conceptualization; Resources; Data curation; Formal analysis; Supervision; Validation; Visualization Writing—original draft; Writing—review and editing. **Anna Moroni**: Conceptualization; Resources; Data curation; Formal analysis; Supervision; Funding acquisition; Writing—original draft; Writing—review and editing.

## Disclosure and competing interests statement

The authors declare no competing interests.

