## [Peer Review File · EMBO Reports]

Modular architecture of K⁺ channels: the functional plasticity of the pore module

Anna Moroni, Oliver Rauh, Tobias Schulze, James VanEtten, and Gerhard Thiel

Corresponding author(s): Anna Moroni (anna.moroni@unimi.it)

Review Timeline:

Submission Date:	10th Feb 25
Editorial Decision:	25th Mar 25
Revision Received:	4th Apr 25
Accepted:	2nd Jun 25

Transaction Report:

Dear Anna,

Thank you for the submission of your review to EMBO reports. Three referees agreed to review your manuscript. So far, we have received two referee reports that are copied below. Since both referees are very positive about your review article and support publication after a few minor concerns are addressed, I have decided to proceed with these two reports.

Please revise your article along the lines suggested by the referees and please provide a detailed point-by-point response to the referee suggestions to facilitate the evaluation of your review. Your revised review will be editorially evaluated and edited in-house to improve clarity, if needed.

When submitting your revised manuscript, we will require a Microsoft Word file (.docx) of the revised manuscript text (with marked track changes) including the figure legends, but without the figures. Please provide these as individual files as .eps, .tif, or .jpg.

There are also a few editorial points that I kindly ask you to address:

- We need a short paragraph called "Box 1: In need of answers", that allows you to describe open questions in the field in the form of a bullet points list.
- Please remove the Author contributions from the manuscript text and make sure that the author contributions in our online manuscript tracking system are correct and up-to-date. The information you specified in the system will be automatically retrieved and typeset into the article. You can enter additional information in the free text box provided, if you wish.
- Please reduce the number of keywords to five.
- Our reviews contain usually 5 to 6 figures. Is it possible to reduce the number by combining two figures? But I am happy to discuss this further.

As for timing, I would like to pencil your review in for our June issue, for which I would need your review back by 20th of April. If you anticipate a problem meeting this deadline, then please let me know and we can aim for another issue.

Thank you for writing this article for us. I look forward to seeing a revised version of your manuscript when it is ready.

Please let me know if you have questions or comments regarding the revision.

Kind regards,

Martina

=====

Referee #2:

The review by Moroni and colleagues is a well-written summarizing review focusing especially on the evolutionary aspects of potassium channels. The argument of this review is of interest and may give a new insight into the study of natural and synthetic ion channels. This timely review addresses also open questions in the field.

This reviewer has only a few comments to further ameliorate the review, specifically the suitability to the wide readership of EMBO Reports.

Several concepts are used in the review that may not be easy to understand for a non-electrophysiologist reader. It would be useful for example to show in a figure and to clearly state the differences between gating current and the current flowing through the pore in order to illustrate and render the concept easily understandable also to non-specialized readers. Definitions of gating charge and tail current should be also given. Likewise, a more detailed description (or a figure) of "swapped" and "non-swapped" architecture could be useful. Some methodological aspect allowing distinction of the diverse types of current mentioned in the review could also be addressed.

Please discuss if the mentioned synthetic channels may have some utility from a translational point of view.

Please mention the proposed/hypothesized function of viral K⁺ channels and give some information about the occurrence of such channels among different types of viruses, in order to underline the relevance of viral channels. Do for example COVID viruses encode such channels? Do these viral K⁺ channels have a preferential location within the host cell?

Minor comments:

Please write out what TEA stays for.

The voltage step protocols used should be shown for the current traces shown.

Please define what LOV domain stays for in Figure 7E.

Referee #3:

Rauh and colleagues have written a great review about the modularity of cation channels that provides a valuable perspective on their evolution, functional mechanism and redesign to have novel properties, including as reporters or actuators of neural signals. The writing is clear, remarkably concise, given the enormous literature, and a strong complement to other reviews of the field. It is welcome too because, despite a history of study, the relation between viral channels and the more familiar bacterial and eucaryotic channels is still known to too few researchers.

The only aspect of the paper that could be boosted is a more concrete description of the mechanics of coupling of domains. The paper talks about how domains can be stand-alone or combined and how association enables one domain control another, but this is very abstract. It would help to give examples of how rearrangements in one domain induce or bias rearrangement in another domain. This, after all, has to be taken into account when one engineers new combinations.

The review is strong and worthy of publication as is. I would just recommend to the authors to consider the following points.

A few suggestions:

1) The VSD alone case is a bit mysterious. Authors refer to finding that one from a Kv becomes a cation selective channel when the pore domain is removed. Can we learn something from why it is cation selective or from what prevents it from functioning this way when the channel is intact? And this is similar to the proton channel. But how can either of these gate and form pores?

2) The authors state that "protein domains with the same architecture of the VSD in Kv channels can be found in various organisms as stand-alone proteins (Fig. 2i) where they function in a tetrameric arrangement as H⁺ conducting channels (Ramsey et al., 2006, Sasaki et al., 2006)." These are dimers not tetramers. And, unlike the Kvs, which are obligate tetramers, these can function as monomers when one breaks the coiled coil C-terminal that holds them together, although at the expense of some cooperativity.

3) There is not mention of the stoichiometry of CiVSP. In glutamate receptors the clamshell ligand binding domains directly interact and must be dimers to exert force, although one can make reporters like GluSnFR that, presumably, work as monomers. But VSDs do not contact each other, so this would seem to lend itself to independent function as monomers, no? Authors could consider a top or bottom view cartoon to complement the topology models to show how the parts attach.

4) "The Ci-VSP element moves in the membrane in response to changes in the electrical field..." did you mean to say 'voltage sensing element' or 'voltage sensor?'

5) It would be helpful to add something more on how molecular motions in one domain couple to / control those of another domain (e.g. voltage sensor to gate; CNBD or Ca²⁺ binding site to gate). Along these lines, how does a peculiar arrangement of the VSD at the C-terminal end of the 14 TM solute transporter enable coupling when the "moving part" is S4?

6) "This difference in pharmacology could however be..." Commas around "however."

7) "In this case voltage sensitivity" case followed by comma.

8) "Some K V family members respond to voltage in the opposite way. They are closed at depolarized voltages and open in a slow manner upon hyperpolarization." Worth mentioning that, at least in HCN as shown by Larsson, S4 still moves outward with depolarization, so this is reverse coupling not reverse voltage sensor motion.

9) Description of inward rectification of Kirs would be clearer if authors explained that blockers are flushed out by inward current.

Dear Martina,

Thanks for sending us the very positive echo on our review manuscript. Since the required changes were not many and very constructive, we have already completed the revision including your editorial requests as well as the changes in the manuscript suggested by the reviewers. Below please find a list of our responses to the requested changes. The modifications are indicated by marked track changes in the manuscript.

We hope that our changes in the manuscript are suitable and make it acceptable for EMBO Reports.

Sincerely yours
Anna

Editorial:

- We need a short paragraph called "Box 1: In need of answers", that allows you to describe open questions in the field in the form of a bullet points list.

Ours response: We have completed this according to the style of published reviews in the journal. See attached file.

- Please remove the Author contributions from the manuscript text and make sure that the author contributions in our online manuscript tracking system are correct and up-to-date. The information you specified in the system will be automatically retrieved and typeset into the article. You can enter additional information in the free text box provided, if you wish.

Ours response: This has been done

- Please reduce the number of keywords to five.

Ours response: This has been done

- Our reviews contain usually 5 to 6 figures. Is it possible to reduce the number by combining two figures? But I am happy to discuss this further.

Ours response: We combined Figs. 4 and 5

Referee #2:

A) Several concepts are used in the review that may not be easy to understand for a non-electrophysiologist reader. It would be useful for example to show in a figure and to clearly state the differences between gating current and the current flowing through the pore in order to illustrate and render the concept easily understandable also to non-specialized readers.

Ours response: We have added at the first mentioning of gating currents on page 5 a layman explanation for the gating currents. From this description it is obvious that this is an intra-protein charge display and not a current through the channel pore.

B) Definitions of gating charge and tail current should be also given.

Ours response: We are now providing this information for the gating charge on page 16. The tail currents are explained together with their measurement in the legend of Fig. 4.

C) Likewise, a more detailed description (or a figure) of "swapped" and "non-swapped" architecture could be useful.

Ours response: We have extended the description of swapped versus non-swapped architecture in the text (page 3). Since the information is not that crucial for the present review, we don't think that it warrants a figure.

D) Some methodological aspect allowing distinction of the diverse types of current mentioned in the review could also be addressed.

Ours response: We have specified throughout the text methodological details like single channel versus macroscopic current measurements and planar lipid bilayer reconstitutions versus whole cell patch clamp recordings. This should provide the desired information.

E) Please discuss if the mentioned synthetic channels may have some utility from a translational point of view.

Ours response: We have added some general sentences on page 24 suggesting the potential function of synthetic channels in nano-sensor devices or even in medicine.

F) Please mention the proposed/hypothesized function of viral K⁺ channels and give some information about the occurrence of such channels among different types of viruses, in order to underline the relevance of viral channels. Do for example COVID viruses encode such channels? Do these viral K⁺ channels have a preferential location within the host cell?

Ours response: Viral ion channels are a very interesting subject but also a very big field. An excursion into the presence of channels in viruses and on their function in infection/replication would be out of the focus of the present review. We have however added on page 8 a few sentences and references, which guide the interested reader to the most relevant reviews in the field.

Minor comments:

Please write out what TEA stays for.

The voltage step protocols used should be shown for the current traces shown.

Please define what LOV domain stays for in Figure 7E.

Ours response: All three comments have been taken care of.

Referee #3:

1) The VSD alone case is a bit mysterious. Authors refer to finding that one from a Kv becomes a cation selective channel when the pore domain is removed. Can we learn something from why it is cation selective or from what prevents it from functioning this way when the channel is intact? And this is similar to the proton channel. But how can either of these gate and form pores?

Ours response: These are indeed interesting questions, for which we cannot provide clear cut answers. But we have expanded the text on page 7 in which we present some educated speculations on these issues in the context of publications on Hv and Kv channels.

2) The authors state that "protein domains with the same architecture of the VSD in Kv channels can be found in various organisms as stand-alone proteins (Fig. 2i) where they function in a tetrameric arrangement as H⁺ conducting channels (Ramsey et al., 2006, Sasaki et al., 2006)." These are dimers not tetramers. And, unlike the Kvs, which are obligate tetramers, these can function as monomers when one breaks the coiled coil C-terminal that holds them together, although at the expense of some cooperativity.

Ours response: Thanks for pointing this out; we have corrected this mistake.

3) There is not mention of the stoichiometry of CiVSP. In glutamate receptors the clamshell ligand binding domains directly interact and must be dimers to exert force, although one can make reporters like GluSnFR that, presumably, work as monomers. But VSDs do not contact each other, so this would seem to lend itself to independent function as monomers, no? Authors could consider a top or bottom view cartoon to complement the topology models to show how the parts attach.

Ours response: We appreciate this comment, and we have on page 5 expanded the text with information on the oligomeric state of the voltage sensing domains in the phosphatase versus in Kv channels. We have pick this point up again on page 22 in the context of the Kv_{syn1} channel.

4) "The Ci-VSP element moves in the membrane in response to changes in the electrical field..." did you mean to say 'voltage sensing element' or 'voltage sensor?'

Ours response: We have corrected this.

5) It would be helpful to add something more on how molecular motions in one domain couple to / control those of another domain (e.g. voltage sensor to gate; CNBD or Ca²⁺ binding site to gate).

Ours response: This is an interesting question and one of our main research interests in the context of HCN channel gating. But in the present review, which is mostly based on a bottom up approach, we have very little on offer to tackle this question. The only information that we can provide comes from the impact of different linkers on the rectification of the synthetic outward rectifier Kcv_{syn1}. We have expanded this chapter on page 22 by reporting data on the impact of linker length and amino acid sequence on rectification of Kcv_{syn1}. We come shortly back to this point later in the context of the light gated BLINK1 channel.

Along these lines, how does a peculiar arrangement of the VSD at the C-terminal end of the 14 TM solute transporter enable coupling when the "moving part" is S4?

Ours response: This was also for us a puzzling question when we became aware of the topology of this protein. In preparation of this response we discovered a recent cryEM structure in this protein. We have added on page 5 some information on the 3D topology of the VSD and the cAMP binding site and on how voltage and cAMP may modulate activity in this protein.

6) "This difference in pharmacology could however be..." Commas around "however."

Ours response: corrected

7) "In this case voltage sensitivity" case followed by comma.

Ours response: corrected

8) "Some K V family members respond to voltage in the opposite way. They are closed at depolarized voltages and open in a slow manner upon hyperpolarization." Worth mentioning that, at least in HCN as shown by Larsson, S4 still moves outward with depolarization, so this is reverse coupling not reverse voltage sensor motion.

Ours response: We have added on page 15 this information. In fact, our demonstration of a slow activating inward rectifying Kcv channel is interesting in the context of some more recent data from the Larsson lab. They show that the direction of rectification is decided at the level of a contact between S4 and the pore at the level of the inner gate. This is in the same region in which mutations in Kcv cause inward rectification. We have added this information

9) Description of inward rectification of Kirs would be clearer if authors explained that blockers are flushed out by inward current.

Ours response: We have added a better description as suggested on page 17

Prof. Anna Moroni
Università degli Studi di Milano
Biosciences
Via Celoria 26
Milan, Mi 20133
Italy

Dear Anna,

I am pleased to inform you that your review article has been accepted for publication in EMBO reports. It was a pleasure to work with you on this review and my congratulations to a review article that will be of great interest for the field.

The figures will be redrawn by a team of graphics designers who will contact you once the first draft is ready for you to review. Once you have approved the figures, I will upload them for you and we will proceed with publication.

Your manuscript will then be processed for publication by EMBO Press. It will be copy edited and you will receive page proofs prior to publication.

When you receive an email from Springer Nature asking you to sign your license agreement, please enter the following code on the payment screen which should remove any charges due: code unavailable.

Should you experience any difficulty, please email publishing@embo.org.

If you have any questions, please do not hesitate to contact me. Thank you for your contribution to EMBO Reports.

Kind regards,

Martina
